# Crowdsourced Clustering: Querying Edges vs Triangles

**Ramya Korlakai Vinayak**
Department of Electrical Engineering
Caltech, Pasadena
ramya@caltech.edu

**Babak Hassibi**
Department of Electrical Engineering
Caltech, Pasadena
hassibi@systems.caltech.edu

## Abstract

We consider the task of clustering items using answers from non-expert crowd workers. In such cases, the workers are often not able to label the items directly, however, it is reasonable to assume that they can compare items and judge whether they are similar or not. An important question is what queries to make, and we compare two types: random edge queries, where a pair of items is revealed, and random triangles, where a triple is. Since it is far too expensive to query all possible edges and/or triangles, we need to work with partial observations subject to a fixed query budget constraint. When a generative model for the data is available (and we consider a few of these) we determine the cost of a query by its entropy; when such models do not exist we use the average response time per query of the workers as a surrogate for the cost. In addition to theoretical justification, through several simulations and experiments on two real data sets on Amazon Mechanical Turk, we empirically demonstrate that, for a fixed budget, triangle queries uniformly outperform edge queries. Even though, in contrast to edge queries, triangle queries reveal dependent edges, they provide more reliable edges and, for a fixed budget, many more of them. We also provide a sufficient condition on the number of observations, edge densities inside and outside the clusters and the minimum cluster size required for the exact recovery of the true adjacency matrix via triangle queries using a convex optimization-based clustering algorithm.

## 1 Introduction

Collecting data from non-expert workers on *crowdsourcing* platforms such as Amazon Mechanical Turk, Zooinverse, Planet Hunters, etc. for various applications has recently become quite popular. Applications range from creating a labeled dataset for training and testing supervised machine learning algorithms [1, 2, 3, 4, 5, 6] to making scientific discoveries [7, 8]. Since the workers on the crowdsourcing platforms are often non-experts, the answers obtained will invariably be noisy. Therefore the problem of designing queries and inferring quality data from such non-expert crowd workers is of great importance.

As an example, consider the task of collecting labels of images, e.g, of birds or dogs of different kinds and breeds. To label the image of a bird, or dog, a worker should either have some expertise regarding the bird species and dog breeds, or should be trained on how to label each of them. Since hiring experts or training non-experts is expensive, we shall focus on collecting labels of images through image comparison followed by clustering. Instead of asking a worker to label an image of a bird, we can show her two images of birds and ask: "Do these two birds belong to the same species?"(Figure 1(a)). Answering this comparison question is much easier than the labeling task and does not require expertise or training. Though different workers might use different criteria for comparison, e.g, color of feathers, shape, size etc., the hope is that, averaged over the crowd workers, we will be able to reasonably resolve the clusters (and label each).

Consider a graph of $n$ images that needs to be clustered, where each pairwise comparison is an 'edge query'. Since the number of edges grows as $\mathcal{O}(n^2)$, it is too expensive to query all edges. Instead, we want to query a subset of the edges, based on our total query budget, and cluster the resulting

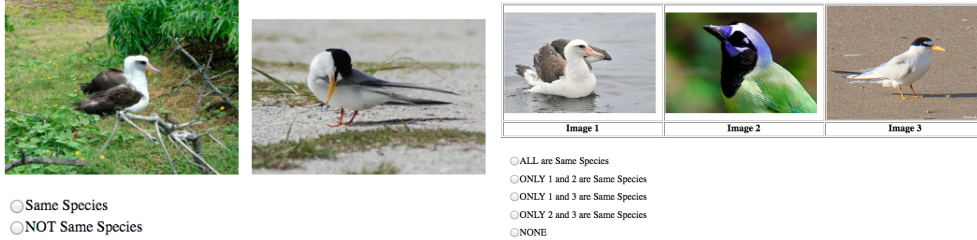

(a) Do these two birds belong to the same species? (b) Which of these birds belong to the same species?

Figure 1: Example of (a) an edge query and (b) a triangle query.

partially observed graph. Of course, since the workers are non-experts, their answers will be noisy and this should be taken into consideration in designing the queries. For example, it is not clear what the best strategy to choose the subsets of edges to be queried is.

## 1.1 Our Contribution

In this work we compare two ways of partially observing the graph: random edge queries, where a pair of items is revealed for comparison, and random triangle queries, where a triplet is revealed. We give intuitive generative models for the data obtained for both types of queries. Based on these models we determine the *cost* of a query to be its entropy (the information obtained from the response to the query). On real data sets where such a generative model may not be known we use the average response time per query as a surrogate for the cost of the query. To fairly compare the use of edge vs. triangle queries we fix the total budget, defined as the (aforementioned) cost per query times the total number of queries. Empirical evidence, based on extensive simulations, as well as two real data sets (images of birds and dogs, respectively), strongly suggests that, for a fixed query budget, querying for triangles significantly outperforms querying for edges. Even though, in contrast to edge queries that give information on *independent edges*, triangle queries give information on *dependent edges*, i.e., edges that share vertices, we (theoretically and empirically) argue that triangle queries are superior because (1) they allow for far more edges to be revealed, given a fixed query budget, and (2) due to the self-correcting nature of triangle queries, they result in much more reliable edges.

Furthermore, for a specific convex optimization-based clustering algorithm, we also provide theoretical guarantee for the exact recovery of the true adjacency matrix via random triangle queries, which gives a sufficient condition on the number of queries, edge densities inside and outside the clusters and the minimum cluster size. In particular, we show that the lower bound of $\Omega(\sqrt{n})$ on the cluster size still holds even though the edges revealed via triangle queries are not independent.

## 1.2 Problem Setup

Consider $n$ items with $K$ disjoint classes/clusters plus outliers (items that do not belong to any clusters). Consider a graph with these $n$ items as nodes. In the true underlying graph $\mathcal{G}^*$, all the items in the same cluster are connected to each other and the items that are not in the same cluster are not connected to each other. We do not have access to $\mathcal{G}^*$. Instead we have a crowdsourced query mechanism that can be used to observe a *noisy* and *partial* snapshot $\mathcal{G}^{\text{obs}}$ of this graph. Our goal is to find the cluster assignments from $\mathcal{G}^{\text{obs}}$. We consider the following two querying methods:

**Random Edge Query:** We sample $E$ edges uniformly at random from $\binom{n}{2}$ possible edges. Figure 1(a) shows an example of an edge query. For each edge observation, there are two possible configurations: (1) Both items are similar, denoted by $ll$, (2) The items are not similar, denoted by $lm$.

**Random Triangle Query:** We sample $T$ triangles uniformly at random from $\binom{n}{3}$ possible triangles. Figure 1(b) shows an example of a triangle query. For each triangle observation, there are five possible configurations (Figure 2):(1) All items are similar, denoted by $lll$, (2) Items 1 and 2 are similar, denoted by $llm$, (3) Items 1 and 3 are similar, denoted by $lml$, (4) Items 2 and 3 are similar, denoted by $mll$, (5) None are similar, denoted by $lmj$.

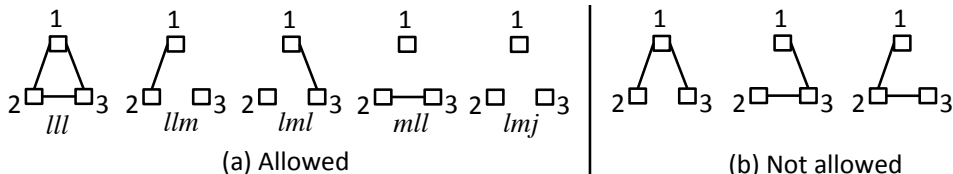

Figure 2: Configurations for a triangle query that are (a) observed and (b) not allowed.

| $\mathbf{Pr}(y\|x)$ | $lll$ | $llm$ | $lmj$ |
|---|---|---|---|
| $lll$ | $p^3 + 3p^2(1-p)$ | $pq^2$ | $q^3$ |
| $llm$ | $p(1-p)^2$ | $p(1-q)^2 + (1-p)q^2 + 2pq(1-q)$ | $q(1-q)^2$ |
| $lml$ | $p(1-p)^2$ | $(1-p)q(1-q)$ | $q(1-q)^2$ |
| $mll$ | $p(1-p)^2$ | $(1-p)q(1-q)$ | $q(1-q)^2$ |
| $lmj$ | $(1-p)^3$ | $(1-p)(1-q)^2$ | $(1-q)^3 + 3q^2(1-q)$ |

Table 1: Query confusion matrix for the triangle block model for the homogeneous case.

## 1.3 Related Works

[9, 10, 11, 12, 13, 14] and references therein focus on the problem of inferring true labels from crowdsoruced multiclass labeling. The common setup in these problems is as follows: A set of items are shown to workers and labels are elicited from them. Since the workers give noisy answers, each item is labeled by multiple workers. Algorithms based on Expectation-Maximization [14] for maximum likelihood estimation and minimax entropy based optimization [12] have been studied for inferring the underlying true labels. In our setup we do not ask the workers to label the items. Instead we use comparison between items to find the clusters of items that are similar to each other.

[15] considers the problem of inferring the complete clustering on $n$ images from a large set of clustering on smaller subsets via crowdsourcing. Each HIT (Human Intelligent Task) is designed such that all of them share a subset of images to ensure overlapping. Each HIT has $M$ images and all the $\binom{M}{2}$ comparisons are made. Each HIT is then assigned to multiple workers to get reliable answers. These clustering are then combined using an algorithm based on variational Bayesian inference. In our work we consider a different setup, where either pairs or triples of images are compared by the crowd to obtain a partial graph on the images which can be clustered.

[16] considers a convex approach to graph clustering with partially observed adjacency matrices, and provides an example of clustering images by crowdsourcing pairwise comparisons. However, it does not consider other types of querying such as triangle queries. In this work, we extend the analysis in [16] and show that similar performance guarantee holds for clustering via triangle queries.

Another interesting line of work is learning embeddings and kernels through triplet comparison tasks in [17, 18, 19, 20, 21, 22] and references therein. The 'triplet comparison' task in these works is of type: 'Is $a$ closer to $b$ or to $c$?', with two possible answers, to judge the relative distances between the items. On the other hand, a triangle query in our work has five possible answers (Figure 1(b)) that gives a clustering (discrete partitioning) of the three items.

## 2 Models

Probability of observing a particular configuration $y$ is given by: $\mathbf{Pr}(y) = \sum_{x \in \mathcal{X}} \mathbf{Pr}(y\|x)\mathbf{Pr}(x)$, where $x$ is the true configuration and $\mathcal{X}$ is the set of true configurations. Let $\mathcal{Y}$ be the set of all observed configurations. Each query has a $|\mathcal{Y}| \times |\mathcal{X}|$ *confusion matrix* $[\mathbf{Pr}(y\|x)]$ associated to it. Note that the columns of this confusion matrix sum to 1, i.e $\sum_{y \in \mathcal{Y}} \mathbf{Pr}(y\|x) = 1$.

### 2.1 Random Edge Observation Models

For the random edge query case, there are two observation configurations, $\mathcal{Y} = \{ll, lm\}$ where $lm$ denotes 'no edge' and $ll$ denotes 'edge'.

**One-coin Edge Model:** Assume all the queries are equally hard. Let the $\zeta$ be the probability of answering a question wrong. Then $\mathbf{Pr}(ll\|ll) = \mathbf{Pr}(lm\|lm) = 1 - \zeta$, $\mathbf{Pr}(lm\|ll) = \mathbf{Pr}(ll\|lm) = \zeta$. This model is inspired by the one-coin Dawid-Skene Model [23], which is used in inference for item label elicitation tasks. This is a very simple model and does not capture the difficulty of a query depending on which clusters the items in the query belong to. In order to incorporate these differences we consider the popular Stochastic Block model (SBM) [24, 25] which is one of the most widely used model for graph clustering.

**Stochastic Block Model (SBM):** Consider a graph on $n$ nodes with $K$ disjoint clusters and outliers. Any two nodes $i$ and $j$ are connected (independent of other edges) with probability $p$ if they belong to the same cluster and with probability $q$ otherwise. That is, $\mathbf{Pr}(ll\|ll) = p$, $\mathbf{Pr}(lm\|ll) = 1 - p$, $\mathbf{Pr}(ll\|lm) = q$ and $\mathbf{Pr}(lm\|lm) = 1 - q$. We assume that the density of the edges inside the clusters is higher than that between the clusters, that is, $p > q$.

### 2.2 Random Triangle Observation Models

For the triangle query model, there are five possible observation configurations (Figure 2), $\mathcal{Y} = \{lll, llm, lml, mll, lmj\}$.

**One-coin Triangle Model:** Let each question be answered correctly with probability $1 - \zeta$, and

| $\mathbf{Pr}(y|x)$ | $lll$ | $llm$ | $lmj$ |
|---|---|---|---|
| $lll$ | $p^3/z_{lll}$ | $pq^2/z_{llm}$ | $q^3/z_{lmj}$ |
| $llm$ | $p(1-p)^2/z_{lll}$ | $p(1-q)^2/z_{llm}$ | $q(1-q)^2$ |
| $lml$ | $p(1-p)^2/z_{lll}$ | $(1-p)q(1-q)/z_{llm}$ | $q(1-q)^2/z_{lmj}$ |
| $mll$ | $p(1-p)^2/z_{lll}$ | $(1-p)q(1-q)/z_{llm}$ | $q(1-q)^2/z_{lmj}$ |
| $lmj$ | $(1-p)^3/z_{lll}$ | $(1-p)(1-q)^2/z_{llm}$ | $(1-q)^3/z_{lmj}$ |

Table 2: Query confusion matrix for the conditional block model for the homogeneous case.

when wrongly answered, all the other configurations are equally confusing. So, $\mathbf{Pr}(lll|lll) = 1 - \zeta$ and $\mathbf{Pr}(llm|lll) = \mathbf{Pr}(lml|lll) = \mathbf{Pr}(mll|lll) = \mathbf{Pr}(lmj|lll) = \zeta/4$ and so on. This model, as in the case of the one-coin model for edge query, does not capture the differences in difficulty for different clusters. In order to include the differences in confusion between different clusters, we consider the following observation models for a triangle query.

For these 3 items in the triangle query, the edges are first generated from the SBM. This can give rise to 8 configurations, out of which 5 are allowed as an answer to triangle query while the rest 3 are not allowed (Figure 2). The two models differ in how they handle the configurations that are not allowed, and are described below:

**Triangle Block Model (TBM):** In this model we assume that a triangle query helps in correctly resolving the configurations that are not allowed. So, when the triangle generated from the SBM takes one of the 3 non-allowed configurations, it is mapped to the true configuration. This gives a $5 \times 5$ query confusion matrix which is given in Table 1. Note that the columns for $lml$ and $mll$ can be filled in a similar manner to that of $llm$.

**Conditional Block Model (CBM):** In this model when a non-allowed configuration is encountered, it is redrawn again. This is equivalent to conditioning on the allowed configurations. Define the normalizing factors, $z_{lll} := 3p^3 - 3p^2 + 1$, $z_{llm} := 3pq^2 - 2pq - q^2 + 1$, $z_{llm} := 3q^3 - 3q^2 + 1$. The $5 \times 5$ query confusion matrix which is given in Table 2.

**Remark:** Note that the SBM (and hence the derived models) can be made more general by considering different edge probabilities $P_{ii}$ for cluster $i$ and $P_{ij} = P_{ji}$ between clusters $i \neq j$.

Some intuitive properties of the triangle query models described in this section are:
1. If $p > q$, then the diagonal term will dominate any other term in a row. That is $\mathbf{Pr}(lll|lll) > \mathbf{Pr}(lll|\star \neq lll), \mathbf{Pr}(llm|llm) > \mathbf{Pr}(llm|\star \neq llm)$ and so on.
2. If $p > 1/2 > q$, then the diagonal term will dominate the other terms in the column, i.e, $\mathbf{Pr}(lll|lll) > \mathbf{Pr}(llm|lll) = \mathbf{Pr}(lml|lll) = \mathbf{Pr}(mll|lll) > \mathbf{Pr}(lmj|lll)$ etc.
3. When there is a symmetry between the items, the observation probability should be the same. That is, if the true configuration is $llm$, then observing $lml$ and $mll$ should be equally likely as item1 and item2 belong to the same cluster and so on. This property will hold good in the general case as well except for when the true configuration is $lmj$. In this case, the probability of observing $llm$, $lml$ and $mll$ can be different as it depends on the clusters to which items 1, 2 and 3 belong.

### 2.3 Adjacency Matrix: Edge Densities and Edge Errors

The adjacency matrix, $\mathbf{A} = \mathbf{A}^T$ of a graph can be partially filled by querying a subset of edges. Since we query edges randomly, most of the edges are seen only once. Some edges might get queried multiple times, in which case, we randomly pick one of them. Similarly we can also partially fill the adjacency matrix from triangle queries. We fill the unobserved entries of the adjacency matrix with zeros. We can perform clustering on $\mathbf{A}$ to obtain a partition of items. The true underlying graph $\mathcal{G}^*$ has perfect clusters (disjoint cliques). So, the performance of clustering on $\mathbf{A}$ depends on how noisy it is. This in turn depends on the probability of error for each revealed edge in $\mathbf{A}$, i.e, what is the probability that a true edge was registered as no-edge and vice versa. The hope is that triangle queries help workers to resolve the edges better and hence have less errors among the revealed edges than those obtained from edge queries.

If we make E edge queries, then the probability of observing an edge is, $r = E/\binom{n}{2}$. If we make $T$ triangle queries, the probability of observing an edge is $r_T = 3T/\binom{n}{2}$. Let $rp$ ($r_T p_T$) and $rq$ ($r_T q_T$) be the edge probability in side the clusters and between the clusters respectively, in $\mathbf{A}$ which is *partially* filled via edge (triangle) queries. For simplicity consider a graph with $K$ clusters of size $m$ each ($n = Km$). The probability that a randomly chosen edge in $\mathbf{A}$ filled via edge query is in error can be computed as: $p_{err}^{edge} := (1 - rp)\,(m - 1)/(n - 1) + rq\,(n - m)/(n - 1)$. Similarly, we can write $p_{err}^{\Delta}$. Under reasonable conditions on the parameters involved, $p_{err}^{\Delta} < p_{err}^{edge}$.

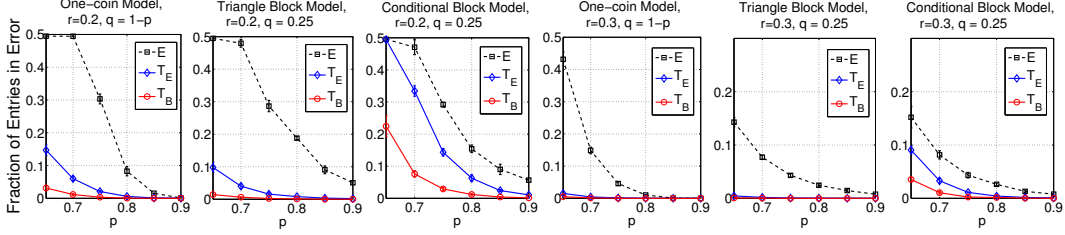

Figure 3: Fraction of entries in error in the matrix recovered via Program 4.1.

For example, in the case of One-coin model, for edge qurey, $rp = r(1 - \zeta)$ and $rq = r\zeta$. For triangle query, $r_T p_T = r_T(1 - 3\zeta/4)$ and $r_T q_T = r_T \zeta/2$. If $r_T < 2r$, we have $r_T q_T < rq$ and $r_T p_T > rp$, and hence $p_{err}^{\Delta} < p_{err}^{edge}$.

For the TBM, when $p > 1/2 > q$, with $r < r_T < r/(1 - q)$, we get $r_T p_T > rp$ and $r_T q_T < rq$, and hence $p_{err}^{\Delta} < p_{err}^{edge}$. For the CBM, when $p > 1/2 > q$, under reasonable assumptions on $r$, $r_T q_T < rq$, but depending on the values of $r$ and $r_T$, $r_T p_T$ can get below $rp$. If the decrease in edge probability between the clusters is large enough to overcome the fall in edge density inside the clusters, then $p_{err}^{\Delta} < p_{err}^{edge}$.

In summary, when $\mathbf{A}$ is filled by triangle queries, the edge density between the clusters decreases and the overall number of edge errors decreases (we observe this in real data as well, see Table 3). Both of these are desirable for clustering algorithms that try to approximate the minimum cut to find the clusters like spectral clustering.

## 3   Value of a Query

To make a meaningful comparison between edge queries and triangle queries, we need to fix a budget. Suppose we have a budget to make $E$ edge queries. To find the number of triangle queries that can be made with the same budget, we need to define the value (cost) of a triangle query. Although a triangle query has 3 edges, they are not independent and hence its relative cost is less than that of making 3 random edge queries. Thus we need a fair way to compare the value of a triangle query to that of an edge query.

Let $\mathbf{s} \in [0,1]^{|\mathcal{Y}|}$, $\sum_{y \in \mathcal{Y}} s_y = 1$ be the probability mass function (pmf) of the observation in a query, with $s_y := \mathbf{Pr}(y) = \sum_{x \in \mathcal{X}} \mathbf{Pr}(y|x)\mathbf{Pr}(x)$. We define the *value* of a query as the information obtained from the observation, which is measured by its *entropy*: $H(\mathbf{s}) = -\sum_{i \in \mathcal{Y}} s_i \log(s_i)$. Ideally, the cost of a query should be proportional to the amount of information it provides. So, if $E$ is the number of edge queries, then the number of triangle queries we can make with the same budget is: $T_B = E \times H_E/H_\Delta$.

We should remark that detetrmining the above cost requires knowledge of the generative model of the graph, which may not be available for empirical data sets. In such situations, a very reasonable cost is the relative time it takes for a worker to respond to a triangle query, compared to an edge query. (In this manner, a fixed budget means a fixed amount of time for the queries to be completed.) A good rule of thumb, which is widely supported by empirical data, is the cost of 1.5, ostensibly because in triangle queries workers need to study three images, rather than two, and so it takes them 50% longer to respond. The end result is that, for a fixed budget, triangle queries reveal twice as many edges.

## 4   Guaranteed Recovery of the True Adjacency Matrix

In this section we provide a sufficient condition for the full recovery of the adjacency matrix corresponding to the underlying true $\mathcal{G}^*$ from partially observed noisy $\mathbf{A}$ filled via random triangle queries. We consider the following convex program from [16]:

$$\underset{\mathbf{L,S}}{\text{minimize}} \ \|\mathbf{L}\|_\star + \lambda\|\mathbf{S}\|_1 \tag{4.1}$$

s. t. $1 \geq \mathbf{L}_{i,j} \geq \mathbf{S}_{i,j} \geq 0$ for all $i,j \in \{1,2,\ldots n\}$, $\mathbf{L}_{i,j} = \mathbf{S}_{i,j}$ whenever $\mathbf{A}_{i,j} = 0$, $\sum_{i,j=1}^{n} \mathbf{L}_{ij} \geq |\mathcal{R}|$

where $\|.\|_\star$ is the nuclear norm (sum of the singular values of the matrix), and $\|.\|_1$ is the $l_1$-norm (sum of absolute values of the entries of the matrix) and $\lambda \geq 0$ is the regularization parameter. $\mathbf{L}$ is the low-rank matrix corresponding to the true cluster structure, $\mathbf{S}$ is the sparse error matrix that accounts only for the missing edges inside the clusters and $|\mathcal{R}|$ is the size of the cluster region.

When $\mathbf{A}$ is filled using a subset of random edge queries, under the SBM with parameters $\{n, n_{min}, K, p, q\}$, [16] provides the following sufficient condition for the guaranteed recovery of the true $\mathcal{G}^*$:

$$n_{\min} \, r \, (p - q) \geq \frac{1}{\lambda} \geq 2\sqrt{n}\sqrt{rq(1 - rq)} + 2\sqrt{n_{\max}}\sqrt{rp(1 - rp) + rq(1 - rq)}, \qquad (4.2)$$

where $n_{\min}$ and $n_{\max}$ are the sizes of the smallest and the largest clusters respectively. We extend the analysis in[16] to the case when $\mathbf{A}$ is filled via a subset of random triangle queries, and obtain the following sufficient condition:

**Theorem 1** *If the following condition holds:*

$$n_{\min} \, r_T \, (p_T - q_T) \geq \frac{1}{\lambda}$$
$$\geq 3 \left( 2\sqrt{n}\sqrt{r_T \frac{q_T}{3}(1 - r_T \frac{q_T}{3})} + 2\sqrt{n_{\max}}\sqrt{r_T \frac{p_T}{3}(1 - r_T \frac{p_T}{3}) + r_T \frac{q_T}{3}(1 - r_T \frac{q_T}{3})} \right)$$

*then Program 4.1 succeeds in recovering the true $\mathcal{G}^*$ with high probability.*

When $\mathbf{A}$ is filled using random edge queries, the entries are independent of each other (since the edges are independent in the SBM). When we use triangle queries to fill $\mathbf{A}$, this no longer holds as the 3 edges filled from a triangle query are not independent. Due to the limited space, we present only the key idea of our proof: The analysis in [16] relies on the independence of entries of $\mathbf{A}$ to use Bernstein-type concentration results for the sum of independent random variables and the bound on the spectral norm of random matrix with independent entries. We make the following observation: Split $\mathbf{A}$ filled via random triangle queries into three parts, $\mathbf{A} = \mathbf{A}_1 + \mathbf{A}_2 + \mathbf{A}_3$. For each triangle query, allocate one edge to each part randomly. If an edge gets queried as a part of multiple triangle queries, keep one of them randomly. Each $\mathbf{A}_i$ now contains independent entries. The edge density in $\mathbf{A}_i$ is $r_T p_T / 3$ and $r_T q_T / 3$ inside the clusters and outside respectively. This allows us to use the results on concentration of sum of independent random variables and the $\mathcal{O}(\sqrt{n})$ bound on the spectral norm of random matrices, with a penalty due to triangle inequality for spectral norm.

It can be seen that, when the number of revealed edges is the same ($r_T = r$) and the probability of correctly identifying edges is the same ($p_T = p$ and $1 - q_T = 1 - q$), then the reovery condition of Theorem 1 is worse than that of (4.2). (This is expected, since triangle queries yield dependent edges.) However, it is overcompensated by the fact that triangle queries result in more reliable edges ($p_T - q_T > p - q$) and also reveal more edges ($r_T > r$, since the relative cost is less than 3).

To illustrate this, consider a graph on $n = 600$ nodes with $K = 3$ clusters of equal size $m = 200$. We generate the adjacency matrices from different models in Section 2 for varying $p$ from 0.65 to 0.9. For the one-coin models, $1 - \zeta = p$. For the rest of the models $q = 0.25$. We run the improved convex program (4.1) by setting $\lambda = 1/\sqrt{n}$. Figure 3 shows the fraction of the entries in the recovered matrix that are wrong compared to the true adjacency matrix for $r = 0.2$ and 0.3 (averaged over 5 runs; $T_E = \lceil E/3 \rceil$ and $T_B = EH_E/H_\Delta$). We note that the error drops significantly when $\mathbf{A}$ is filled via triangle queries than via edge queries.

## 5  Performance of Spectral Clustering: Simulated Experiments

We generate adjacency matrices from the edge query and the triangle query models (Section 2) and run the spectral clustering algorithm [26] on them. We compare the output clustering with the ground truth via *variation of information* (VI) [27] which is defined for two clusterings (partitions) of a dataset and has information theoretical justification. Smaller values of VI indicate a closer match and a VI of 0 means that the clusterings are identical. We compare the performance of the spectral clustering algorithms on the partial adjacency matrices obtained from querying: (1) $E = \lceil r\binom{n}{2} \rceil$ random edges, (2) $T_B = E \times H_E/H_\Delta$ random triangles, which has the same budget as querying $E$ edges and (3) $T_E = \lceil E/3 \rceil < T_B$ random triangles, which has same number of edges as in the adjacency matrix obtained by querying $E$ edges.

**Varying Edge Density Inside the Clusters:** Consider a graph on $n = 450$ nodes with $K = 3$ clusters of equal size $m = 150$. We vary edge density inside the cluster $p$ from 0.55 to 0.9. For the one-coin models, $1 - \zeta = p$, and $q = 0.25$ for the rest. Figure 4 shows the performance of spectral clustering for $r = 0.15$ and $r = 0.3$ (averaged over 5 runs).

**Varying Cluster Sizes:** Let $N = 1200$. Consider a graph with $K$ clusters of equal sizes $m = \lfloor N/K \rfloor$ and $n = K\,m$. We vary $K$ from 2 to 12 which varies the cluster sizes from 600 (large clusters) to 100 (small clusters, note that $\sqrt{1200} \approx 35$). We set $p = 0.7$. For the one-coin models

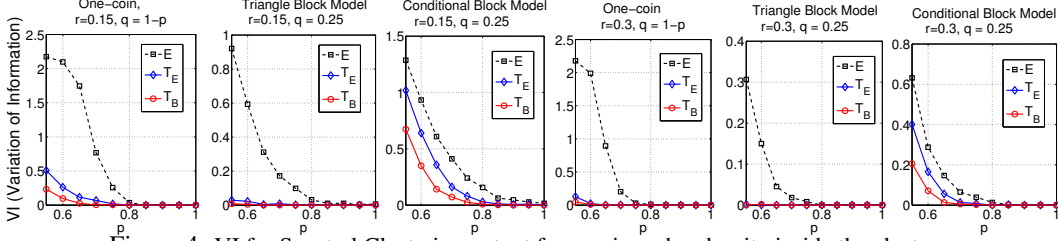

Figure 4: VI for Spectral Clustering output for varying edge density inside the clusters.

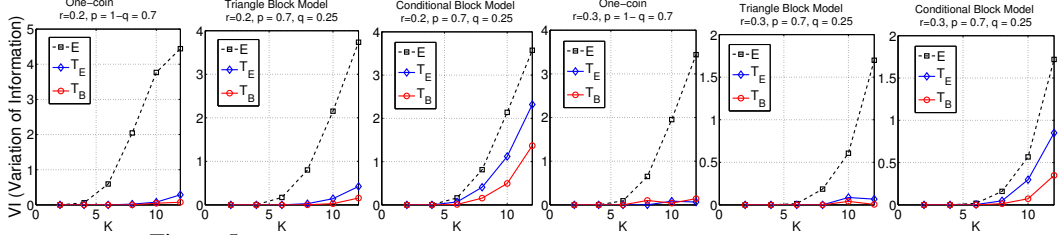

Figure 5: VI for Spectral Clustering output for varying number of clusters (K).

$1 - \zeta = p$ and $q = 0.25$ for the rest. Figure 5 shows the performance of spectral clustering for $r = 0.2$ and $0.3$. The performance is significantly better with triangle queries compared to that with edge queries.

## 6 Experiments on Real Data

We use Amazon Mechanical Turk as crowdsourcing platform. For edge queries, each HIT (Human Intelligence Task) has 30 queries of random pairs, a sample is shown in Figure 1(a). For triangle queries, each HIT has 20 queries, with each query having 3 random images, a sample is shown in Figure 1(b). Each HIT is answered by a unique worker. Note that we do not provide any examples of different classes or any training to do the task. We fill **A** as described in Section 2.3 and run the $k$-means, the Spectral Clustering and Program 4.1 followed by Spectral Clustering on it. Since we do not know the model parameters and hence have no access to the entropy information, we can use the the average time taken as the "cost" or value of the query. For $E$ edge comparisons, the equivalent number of triangle comparisons would be $T = E \times t_E / t_\Delta$, where $t_E$ and $t_\Delta$ are average time taken to answer an edge query and a triangle query respectively. We consider two datasets:

1. **Dogs3** dataset has images of the following 3 breeds of dogs from the Stanford Dogs Dataset [28]: Norfolk Terrier (172), Toy Poodle (150) and Bouvier des Flanders (151), giving a total of 473 dogs images. On an average a worker took $t_E = 8.4s$ to answer an edge query and $t_\Delta = 11.7s$ to answer a triangle query.
2. **Birds5** dataset has 5 bird species from CUB-200-2011 dataset [29]: Laysan Albatross (60), Least Tern (60), Artic Tern (58), Cardinal (57) and Green Jay (57). We also add 50 random species as outliers, giving us a total if 342 bird images. On an average, workers took $t_E = 8.3s$ to answer one edge query and $t_\Delta = 12.1s$ to answer a triangle query.

Details of the data obtained from edge query and triangle query experiments is summarized in Table 3. Note that the error in the revealed edges drop significantly for triangle queries.

For the Dogs3 dataset, the empirical edge densities inside and between the clusters for **A** obtained from the edge queries ($\hat{P}_E$) and the triangle queries ($\hat{P}_T$) is:

$$\hat{P}_E = \begin{bmatrix} 0.7577 & 0.1866 & 0.2043 \\ 0.1866 & 0.6117 & 0.2487 \\ 0.2043 & 0.2487 & 0.7391 \end{bmatrix}, \; \hat{P}_T = \begin{bmatrix} 0.7139 & 0.1138 & 0.1253 \\ 0.1138 & 0.6231 & 0.1760 \\ 0.1253 & 0.1760 & 0.7576 \end{bmatrix}.$$

| E: Edge, T: $\Delta$ | # Workers | # Unique Edges | % of Edges Seen | % of Edge Errors |
|---|---|---|---|---|
| Dogs3, Edge Query | 300 | $E' = 8630$ | 7.73% | 25.2% |
| Dogs3, $\Delta$ Query | 150 | $3T'_E = 8644$ | 7.74% | 19.66% |
| Dogs3, $\Delta$ Query | 320 | $3T' = 17,626$ | 15.79% | 20% |
| Birds5, Edge Query | 300 | $E' = 8319$ | 14.27% | 14.82% |
| Birds5, $\Delta$ Query | 155 | $3T'_E = 8600$ | 14.74% | 10.96% |
| Birds5, $\Delta$ Query | 285 | $3T' = 14,773$ | 25.34% | 11.4% |

Table 3: Summary of the data colleced in the real experiments.

| Query (E: Edge, T: $\Delta$) | k-means | Spectral Clustering | Convex Program |
|---|---|---|---|
| $E' = 8630$ | $0.8374 \pm 0.0121$ (K=2) | $0.6972 \pm 0$ (K = 3) | $0.5176 \pm 0$ (K=3) |
| $3T'_E = 8644$ | $0.6675 \pm 0.0246$ (K=3) | $0.5690 \pm 0$ (K=3) | $0.4605 \pm 0$ (K = 3) |
| $3T' = 17626$ | $\mathbf{0.3268} \pm 0$ (K=3) | $\mathbf{0.3470} \pm 0$ (K=3) | $\mathbf{0.2279} \pm 0$ (K = 3) |

Table 4: VI for clustering output by k-means and spectral clustering for the Dogs3 dataset.

| Query | k-means | Spectral Clustering | Convex Program |
|---|---|---|---|
| $E' = 8319$ | $1.4504 \pm 0.0338$ (K = 2) | $1.2936 \pm 0.0040$ (K = 4) | $1.0392 \pm 0$ (K = 4) |
| $3T'_E = 8600$ | $1.1793 \pm 0.0254$ (K = 3) | $1.1299 \pm 0 (K = 4)$ | $\mathbf{0.9105} \pm 0$ (K=4) |
| $3T' = 14,773$ | $\mathbf{0.7989} \pm 0$ (K = 4) | $\mathbf{0.8713} \pm 0$ (K = 4) | $0.9135 \pm 0$ (K = 4) |

Table 5: VI for clustering output by k-means and spectral clustering for the Birds5 dataset.

For the Birds5 dataset, the empirical edge densities within and between various clusters in $\mathbf{A}$ filled via edge queries ($\hat{P}_E$) and triangle queries ($\hat{P}_T$) are:

$$\hat{P}_E = \begin{bmatrix} 0.801 & 0.304 & 0.208 & 0.016 & 0.032 & 0.100 \\ 0.304 & 0.778 & 0.656 & 0.042 & 0.131 & 0.123 \\ 0.208 & 0.656 & 0.912 & 0.062 & 0.094 & 0.096 \\ 0.016 & 0.042 & 0.062 & 0.855 & 0.154 & 0.110 \\ 0.032 & 0.131 & 0.094 & 0.154 & 0.958 & 0.158 \\ 0.100 & 0.123 & 0.096 & 0.110 & 0.158 & 0.224 \end{bmatrix}, \hat{P}_T = \begin{bmatrix} 0.786 & 0.207 & 0.151 & 0.011 & 0.021 & 0.058 \\ 0.207 & 0.797 & 0.625 & 0.023 & 0.047 & 0.1 \\ 0.151 & 0.625 & 0.865 & 0.024 & 0.06 & 0.071 \\ 0.011 & 0.023 & 0.024 & 0.874 & 0.059 & 0.076 \\ 0.021 & 0.047 & 0.06 & 0.059 & 0.943 & 0.08 \\ 0.058 & 0.1 & 0.071 & 0.078 & 0.08 & 0.182 \end{bmatrix}.$$

As we see the triangle queries give rise to an adjacency matrix with significantly less confusion across the clusters (compare the off-diagonal entries in $\hat{P}_E$ and $\hat{P}_T$).

Tables 4 and 5 show the performance of clustering algorithms (in terms of variation of information) for the two datasets. The no. of clusters found is given in brackets. We note that for both the datasets, the performance is significantly better with triangle queries than with edge queries. Furthermore, even with less triangle queries ($3T'_E \approx E$) than that is allowed by the budget, the clustering obtained is better compared to edge queries.

## 7   Summary

In this work we compare two ways of querying for crowdsourcing clustering using non-experts: random edge comparisons and random triangle comparisons. We provide simple and intuitive models for both. Compared to edge queries that reveal independent entries of the adjacency matrix, triangle queries reveal dependent ones (edges in a triangle share a vertex). However, due to their error-correcting capabilities, triangle queries result in more reliable edges and, furthermore, because the cost of a triangle query is less than that of 3 edge queries, for a fixed budget, triangle queries reveal many more edges. Simulations based on our models, as well as empirical evidence strongly support these facts. In particular, experiments on two real datasets suggests that clustering items from random triangle queries significantly outperforms random edge queries when the total query budget is fixed. We also provide theoretical guarantee for the exact recovery of the true adjacency matrix using random triangle queries. In the future we will focus on exploiting the structure of triangle queries via tensor representations and sketches, which might further improve the clustering performance.

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
