[Supplementary Material]

# Supplementary Material

# Contents

# 1  Some Notes On Real Data Experiments

Tables 1 and 2 has the overall (averaged over all types) empirical observation probability matrix for triangle query for Dogs3 dataset and Birds5 dataset respectively. The column is the true configuration and the rows are the observed configuration.

Note that the diagonal is dominant in both row and column. So,

$$\mathbf{Pr}(kkk|kkk) > \mathbf{Pr}(kkk|\star \neq kkk) \text{ and } \mathbf{Pr}(kkk|kkk) > \mathbf{Pr}(\star \neq kkk|kkk).$$

Further note that $kkk$ is roughly equally confused as $kkl, klk, lkk$. Similarly, $kkl$ is roughly equally confused as $klk$ and $lkk$ etc.

|      | kkk      | kkl      | klk      | lkk      | klj      |
|------|----------|----------|----------|----------|----------|
| kkk  | **0.56** | 0.08     | 0.09     | 0.09     | 0.04     |
| kkl  | 0.13     | **0.60** | 0.04     | 0.05     | 0.11     |
| klk  | 0.12     | 0.03     | **0.58** | 0.04     | 0.1      |
| lkk  | 0.10     | 0.03     | 0.05     | **0.56** | 0.09     |
| klj  | 0.09     | 0.26     | 0.24     | 0.26     | **0.66** |

Table 1: Empirical observation probability matrix for triangle query for Dogs3 dataset.

|      | kkk      | kkl      | klk      | lkk      | klj      |
|------|----------|----------|----------|----------|----------|
| kkk  | **0.79** | 0.1      | 0.09     | 0.08     | 0.02     |
| kkl  | 0.07     | **0.76** | 0.04     | 0.03     | 0.08     |
| klk  | 0.05     | 0.02     | **0.71** | 0.01     | 0.08     |
| lkk  | 0.04     | 0.01     | 0.02     | **0.73** | 0.08     |
| klj  | 0.05     | 0.11     | 0.14     | 0.15     | **0.74** |

Table 2: Empirical observation probability matrix for triangle query for Birds5 dataset.

The empirical observation probability matrix for triangle query for Norfolk Terrier in Dogs3 Dataset is shown in Table 3. We can make similar observation as in case of Table 1.

We have chosen the Birds5 dataset such that there is sufficient confusion. There are two categories, Cardinal (red in color) and Green Jay that are vibrant in color (Figure 2). Whereas, Albatross, Least Tern and Arctic Tern are all white birds (Figure 1). Furthermore, the difference between Least Tern and Arctic Tern is subtle (they have different beak and feet color).

|      | 111 | 112 | 121 | 211 | 113 | 131 | 311 | 123 |
|------|-----|-----|-----|-----|-----|-----|-----|-----|
| kkk  | **0.5710** | 0.0552 | 0.0745 | 0.0950 | 0.0734 | 0.0950 | 0.0988 | 0.0381 |
| kkl  | 0.1081 | **0.6310** | 0.0353 | 0.0329 | **0.6409** | 0.0372 | 0.0277 | 0.0894 |
| klk  | 0.1216 | 0.0241 | **0.6039** | 0.0329 | 0.0193 | **0.6446** | 0.0277 | 0.0970 |
| lkk  | 0.1250 | 0.0241 | 0.0431 | **0.5766** | 0.0232 | 0.0372 | **0.5850** | 0.1136 |
| klj  | 0.0743 | 0.2655 | 0.2431 | 0.2628 | 0.2432 | 0.1860 | 0.2608 | **0.6618** |

Table 3: Empirical observation probability matrix for triangle query for Norfolk Terrier in Dogs3 dataset.

(1)Albatross     (2) Least Tern     (3) Arctic Tern

Figure 1: Sample Images of (1)Albatross, (2) Least Tern and (3) Arctic Tern.

Note that Table 2 is averaged over all the 5 categories plus outliers of birds in the dataset. A subset of columns of empirical observation probability matrix for triangle query for Least Tern in Birds5 Dataset is shown in Table 4. Least Tern is indexed by 2, Arctic Tern by 3 and Albatross by 1. Since Least Tern and Arctic Tern are very close to each other compared to Albatross(Figure 1), there is more confusions between them. So, $kkl$ with Least Tern and Arctic Tern is more likely to get mapped as $kkk$.

|      | 222 | 221 | 212 | 122 | 223 | 232 | 322 |
|------|-----|-----|-----|-----|-----|-----|-----|
| kkk  | **0.770** | 0.219 | 0 | 0.132 | **0.485** | **0.4** | **0.548** |
| kkl  | 0.077 | **0.531** | 0.093 | 0 | 0.242 | 0.171 | 0.032 |
| klk  | 0.077 | 0.094 | **0.688** | 0.053 | 0.121 | 0.229 | 0.129 |
| lkk  | 0.038 | 0 | 0 | **0.71** | 0.061 | 0.057 | 0.161 |
| klj  | 0.038 | 0.156 | 0.219 | 0.105 | 0.091 | 0.143 | 0.130 |

Table 4: Empirical observation probability matrix for triangle query for Least Tern in Birds5 dataset.

**Outlier Detection:** If the output of the convex program has all zero row (or column), then that node is an outlier. Recall that we have 50 outliers in Birds5 dataset. Convex Program was able to find 11 outliers out of these and another from Albatross using data from edge queries. The one picked from Albatross indeed looks weird 3, so it is not unreasonable to expect it as an outlier. From triangle queries, it was able to find 21 out of 50 outliers.

## 2   Edge Density and Edge Errors

If we make E edge queries, then the probability of observing an edge is, $r = E/\binom{n}{2}$. If we make $T$ triangle queries, the probability of observing an edge is $r_t = 3T/\binom{n}{2}$. Let $rp_E$ $(r_t p_T)$ and $rq_E$ $(r_t q_T)$ be the edge probability in side the clusters and between the clusters respectively, in **A** which is *partially* filled via edge (triangle) queries.

For simplicity consider a graph with $K$ clusters of size $m$ each ($n = Km$). The probability that a randomly chosen edge in **A** filled via edge query is in error can be computed as:

$$p_{err}^{edge} := (1 - rp_E)\,(m - 1)/(n - 1) + rq_E\,(n - m)/(n - 1)$$

Similarly, we can write

$$p_{err}^{\Delta} := (1 - r_t p_T)\,(m - 1)/(n - 1) + r_T q_T\,(n - m)/(n - 1)$$

### 2.1   One-coin Model

For edge query: $rp_E = r\,(1 - \zeta)$ and $rq_E = r\,\zeta$.

(4) Cardinal        (5) Green Jay

Figure 2: Sample Images of (4)Cardinal and (5) Green Jay.

Figure 3: Outlier found in Albatross

For triangle query:

$$r_t p_T = r_t \left( \frac{m-2}{n-2}(1 - \zeta + \frac{\zeta}{4}) + \frac{n-m}{n-2}(\frac{\zeta}{4} + 1 - \zeta) \right)$$
$$= r_t \left(1 - \frac{3\zeta}{4}\right)$$

$$r_t q_T = r_t \left( 2\frac{m-1}{n-2}\frac{2\zeta}{4} + \frac{n-2m}{n-2}\frac{2\zeta}{4} \right)$$
$$= r_t \frac{\zeta}{2}$$

43 When $r = r_t$, $q_T = \zeta/2 < q_E$ and $p_T = 1 - 3\zeta/4 > p_E$, and hence $p_{err}^{\Delta} < p_{err}^{edge}$. If $r < r_t < 2r$,
44 we still get $r_t p_T > r p_E$, $r_t q_T < r q_E$ and $p_{err}^{\Delta} < p_{err}^{edge}$.

45 ## 2.2  Triangle Block Model

46 Since TBM is derived from SBM, $r p_E = r\, p$ and $r q_E = r\, q$.

47 Assume $p > 1/2 > q$.

$$r_t p_T = r_t \left( \frac{m-2}{n-2}(p^3 + 3p^2(1-p) + p(1-p)^2) + \frac{n-m}{n-2}(pq^2 + p(1-q)^2 + (1-p)q^2 + 2pq(1-q)) \right)$$
$$= r_t\, p + r_t(1-p)\left( \frac{m-2}{n-2}p^2 + \frac{n-m}{n-2}q^2 \right) > r_t\, p$$

Figure 4: Edge error probability for Conditional Block Model, n = 900 with K = 3 clusters of size m = 300 each.

Figure 5: Edge error probability for Conditional Block Model, n = 1200 with K = 12 clusters of size m = 100 each.

$$
\begin{aligned}
r_t q_T &= \left( 2\frac{m-1}{n-2}(pq^2 + (1-p)q(1-q)) + \frac{n-2m}{n-2}(q^3 + q(1-q)^2) \right) \\
&= r_t\, q \left( 1 - q - (1-2q)\left( 2\frac{m-1}{n-2}p + \frac{n-2m}{n-2}q \right) \right) \\
&< r_t\, q\,(1-q) \quad \text{since } p > 1/2 > q.
\end{aligned}
$$

48  When $r_t = r$, $p_T = p_E +$ positive term $> p$ and $q_T = q_E -$ positive term $< q$, and hence
49  $p_{err}^{\Delta} < p_{err}^{edge}$.

50  If $r < r_t < r/(1-q)$, we get $r_t p_T > r p_E$ and $r_t q_T < r q_E$.

51  ## 2.3 Conditional Block Model

52  Since CBM is derived from SBM, $r p_E = r\, p$ and $r q_E = r\, q$.

Figure 6: Edge error probability for Conditional Block Model, n = 900 with K = 3 clusters of size m = 300 each.

Figure 7: Edge error probability for Conditional Block Model, n = 1200 with K = 12 clusters of size m = 100 each.

$$r_t p_T = \mathbf{Pr}(obs)\left(\mathbf{Pr}(kkk)\mathbf{Pr}(kkk, kkl|kkk) + \mathbf{Pr}(kkl)\mathbf{Pr}(kkk, kkl|kkl)\right)$$

$$= r_t \left(\frac{m-2}{n-2}\frac{p^3 + p(1-p)^2}{3p^3 - 3p^2 + 1} + \frac{n-m}{n-2}\frac{pq^2 + p(1-q)^2}{3pq^2 - 2pq - q^2 + 1}\right)$$

$$= r_t\, p \left(\frac{m-2}{n-2}\frac{1 + 2p^2 - 2p}{3p^3 - 3p^2 + 1} + \frac{n-m}{n-2}\frac{1 + 2q^2 - 2q}{3pq^2 - 2pq - q^2 + 1}\right)$$

$$\approx r_t p \left(\frac{1}{K}\frac{1 + 2p^2 - 2p}{3p^3 - 3p^2 + 1} + (K-1)\frac{1 + 2q^2 - 2q}{3pq^2 - 2pq - q^2 + 1}\right) \quad \text{for large n.}$$

$$r_t q_T = \mathbf{Pr}(obs)\left(\mathbf{Pr}(klk)\mathbf{Pr}(kkk, kkl|klk) + \mathbf{Pr}(kkl)\mathbf{Pr}(kkk, kkl|lkk) + \mathbf{Pr}(klj)\mathbf{Pr}(kkk, kkl|klj)\right)$$

$$= r_t \left(2\frac{m-1}{n-2}\frac{pq^2 + (1-p)q(1-q)}{3pq^2 - 2pq - q^2 + 1} + \frac{n-2m}{n-2}\frac{q^3 + q(1-q)^2}{3q^3 - 3q^2 + 1}\right)$$

$$= r_t q \left(2\frac{m-1}{n-2}\frac{pq + (1-p)(1-q)}{3pq^2 - 2pq - q^2 + 1} + \frac{n-2m}{n-2}\frac{q^2 + (1-q)^2}{3q^3 - 3q^2 + 1}\right)$$

$$= r_t q \left(2\frac{m-1}{n-2}\frac{2pq + 1 - p - q}{3pq^2 - 2pq - q^2 + 1} + \frac{n-2m}{n-2}\frac{2q^2 + 1 - 2q}{3q^3 - 3q^2 + 1}\right)$$

$$\approx r_t\, q \left(\frac{2}{K}\frac{2pq + 1 - p - q}{3pq^2 - 2pq - q^2 + 1} + (K-2)\frac{2q^2 + 1 - 2q}{3q^3 - 3q^2 + 1}\right) \quad \text{for large n.}$$

53   Consider two examples:

54   1. Large Clusters: A graph on $n = 900$ nodes with $K = 3$ clusters of equal size $m = 300$
55   ($m = n/3$).

56   2. Small Clusters: A graph on $n = 1200$ nodes with $K = 12$ clusters of equal size $m = 100$. Note
57   that $\sqrt{1200} \approx 35$. So, $m \approx 3\sqrt{n}$.

58   Figures 4 and 5 show the probability of edge errors for the conditional block model $p_{err}^{\Delta}$ compared to
59   the SBM for edge query, $p_{err}^{edge}$ for $q = 0.15, 0.25$ and $0.35$ for the two cases considered. The first row
60   is when $r_t = r$ and the second row is when $r_t$ is scaled according to entropy, that is, $r_t = 3rH_E/H_\Delta$.
61   Note that $p_{err}^{\Delta} < p_{err}^{edge}$.

62   When we push $q$ closer to $1/2$, we can expect the edge errors in conditional query query model to
63   start getting worse. This is because a lot of triangle queries of the form $kkl, klk, lkk, klj$ are easily
64   confused to $kkk$. So the more queries of these kinds (which are plently, especially when clusters
65   are small) we make, the more edges with errors are obtained. Figures 6 and 7 show the probability
66   of edge errors for the conditional block model $p_{err}^{\Delta}$ compared to the SBM for edge query, $p_{err}^{edge}$ for
67   $q = 0.45, 0.47$ and $0.49$ for the two cases considered. Note that still $p_{err}^{\Delta} < p_{err}^{edge}$ in all except when
68   $q = 0.49$ with small clusters and more edges are revealed.

## 69   3   Proof Details

70   Recall that we use the following Convex Program from [1]:

$$\underset{\mathbf{L},\mathbf{S}}{\text{minimize}}\ \|\mathbf{L}\|_\star + \lambda\|\mathbf{S}\|_1 \tag{3.1}$$

$$\text{s. t. } 1 \geq \mathbf{L}_{i,j} \geq \mathbf{S}_{i,j} \geq 0 \text{ for all } i,j \in \{1, 2, \dots n\},\ \mathbf{L}_{i,j} = \mathbf{S}_{i,j} \text{ whenever } \mathbf{A}_{i,j} = 0,\ \sum_{i,j=1}^{n}\mathbf{L}_{ij} \geq |\mathcal{R}|$$

71   where $\|.\|_\star$ is the nuclear norm (sum of the singular values of the matrix), and $\|.\|_1$ is the $l_1$-norm
72   (sum of absolute values of the entries of the matrix) and $\lambda \geq 0$ is the regularization parameter. $\mathbf{L}$
73   is the low-rank matrix corresponding to the true adjacency matrix, $\mathbf{S}$ is the sparse error matrix that
74   accounts only for the missing edges inside the clusters.

Figure 8: Illustration of $\{\mathcal{R}_{i,j}\}$ dividing $[n] \times [n]$ into disjoint regions similar to a grid

75  The true $\mathcal{G}^*$ is a union of cliques. So, the true adjacency matrix is of the form:

$$\mathbf{L}^0_{i,j} = \begin{cases} 1 & \text{if both } \{i,j\} \in \mathcal{C}_l \text{ for some } l \leq K, \\ 0 & \text{otherwise.} \end{cases} \tag{3.2}$$

76  where $C_l$ denotes the set of nodes that belong to cluster $l$.

77  Note that, $\mathbf{L}^0 = \mathbf{U}\Lambda\mathbf{U}^T$, where $\Lambda = \text{diag}\{n_1, n_2, \ldots, n_K\}$, with $n_i = |\mathcal{C}_i|$ being the size of cluster
78  $i$ and $\mathbf{U} = [\mathbf{u}_1 \ \ldots \ \mathbf{u}_K] \in \mathbb{R}^{n \times K}$,

$$\mathbf{u}_{l,i} = \begin{cases} \frac{1}{\sqrt{n_l}} & \text{if } i \in \mathcal{C}_l \\ 0 & \text{else} \end{cases} \tag{3.3}$$

79  Let:

$$\mathbf{S}^0_{i,j} = \begin{cases} 1 & \text{if both } \{i,j\} \in \mathcal{C}_l \text{ for some } l \leq K, \text{ and } \mathbf{A}_{i,j} = 0, \\ 0 & \text{otherwise.} \end{cases}$$

80  Our claim is that $(\mathbf{L}^0, \mathbf{S}^0)$ is the unique optimal solution to the Program 3.1. Our proof closely
81  follows the analysis in [1].

82  Let $\mathcal{R}_{i,j} = \mathcal{C}_i \times \mathcal{C}_j$ for $1 \leq i, j \leq K + 1$. One can see that $\{\mathcal{R}_{i,j}\}$ divides $[n] \times [n]$ into $(K+1)^2$
83  disjoint regions similar to a grid which is illustrated in the Figure 8. Thus, $\mathcal{R}_{i,i}$ is the region induced
84  by $i$'th cluster for any $1 \leq i \leq K$. Let $\mathcal{A}_1$ denote the entries of $\mathbf{A}$ that are observed to be 1 and $\mathcal{A}_0$
85  are entries of $\mathbf{A}$ that are observed to be 0.

86  It is easy to see that the $(\mathbf{L}^0, \mathbf{S}^0)$ pair is feasible. To show that it is unique optimal solution, we need
87  the following to hold:

$$\|\mathbf{L}^0 + \mathbf{E}^L\|_\star + \lambda \|\mathbf{S}^0 + \mathbf{E}^S\|_1 - (\|\mathbf{L}^0\|_\star + \lambda \|\mathbf{S}^0\|_1) \geq \langle \partial\|\mathbf{L}^0\|_\star, \mathbf{E}^L \rangle + \lambda \langle \partial\|\mathbf{S}^0\|_1, \mathbf{E}^S \rangle > 0,$$

88  for any feasible perturbation $(\mathbf{E}^L, \mathbf{E}^S)$. Note that $\partial\|\mathbf{L}^0\|_\star$ and $\partial\|\mathbf{S}^0\|_1$ are subgradients of nuclear
89  norm and $\ell_1$-norm respectively at the points $(\mathbf{L}^0, \mathbf{S}^0)$.

90  $\partial\|\mathbf{L}^0\|_\star$ is of the form $\mathbf{U}\mathbf{U}^T + \mathbf{W}$ such that $\mathbf{W} \in \mathcal{M}_U := \{\mathbf{X} : \mathbf{X}\mathbf{U} = \mathbf{U}^T\mathbf{X} = 0, \|\mathbf{X}\| \leq 1\}$. and
91  $\partial\|\mathbf{S}^0\|_1$ is of the form $\text{sign}(\mathbf{S}^0) + \mathbf{Q}$ where $\mathbf{Q}_{i,j} = 0$ if $\mathbf{S}^0_{i,j} \neq 0$ and $\|\mathbf{Q}\|_\infty \leq 1$.

$$\|\mathbf{L}^0 + \mathbf{E}^L\|_\star + \lambda \|\mathbf{S}^0 + \mathbf{E}^S\|_1 - (\|\mathbf{L}^0\|_\star + \lambda \|\mathbf{S}^0\|_1) \geq \langle \partial\|\mathbf{L}^0\|_\star, \mathbf{E}^L \rangle + \lambda \langle \partial\|\mathbf{S}^0\|_1, \mathbf{E}^S \rangle$$
$$= \langle \mathbf{U}\mathbf{U}^T + \mathbf{W}, \mathbf{E}^L \rangle + \lambda \langle \text{sign}(\mathbf{S}^0) + \mathbf{Q}, \mathbf{E}^S \rangle$$

92  Note that, $\mathbf{L}_{i,j} = \mathbf{S}_{i,j}$ whenever $\mathbf{A}_{i,j} = 0$. So, $\mathbf{L}^0_{\mathcal{A}_0} = \mathbf{S}^0_{\mathcal{A}_0}$. Further, $\mathbf{S}$ can be split as $\mathbf{S} = \S_1 + \S_2$,
93  where $\S_1$ corresponds to entries that are observed in $\mathbf{A}$ and $\mathbf{S}_{\text{rest}}$ denotes the entries of $\mathbf{S}$ other than
94  those corresponding to the observed entries of $\mathbf{A}$. We note that at the optimal, $\S_2 = 0$, since if
95  otherwise, the objective can be strictly decreased by setting $\mathbf{S}_{\text{rest}} = 0$. So, without loss of generality,
96

$$\mathbf{S}^0 = \mathbf{L}^0_{\mathcal{A}_0}. \tag{3.4}$$

Using $\text{sign}(\mathbf{S}^0) = \mathbb{1}^{n \times n}_{\mathcal{A}_0 \cap \mathcal{R}}$, and choosing $\mathbf{Q} = \mathbb{1}^{n \times n}_{\mathcal{A}_0 - (\mathcal{A}_0 \cap \mathcal{R})}$, we get,

$$\|\mathbf{L}^0 + \mathbf{E}^L\|_\star + \lambda \|\mathbf{S}^0 + \mathbf{E}^S\|_1 - (\|\mathbf{L}^0\|_\star + \lambda \|\mathbf{S}^0\|_1) \geq \langle \mathbf{W}, \mathbf{E}^L \rangle$$
$$+ \underbrace{\sum_{i=1}^{K} \frac{1}{n_i} \text{sum}(\mathbf{E}^L_{R_{i,i}}) + \lambda \left( \text{sum}(\mathbf{E}^L_{\mathcal{A}_0}) \right)}_{:= g(\mathbf{E}^L)}$$

(3.5)

for any $\mathbf{W} \in \mathcal{M}$. Note that $\text{sum}(\mathbf{M})$ denotes sum of all the entries of matrix $\mathbf{M}$.

Define,

$$g(\mathbf{E}) := \sum_{i=1}^{K} \frac{1}{n_i} \text{sum}(\mathbf{E}_{\mathcal{R}_{i,i}}) + \lambda \text{sum}(\mathbf{E}_{\mathcal{A}_0})). \tag{3.6}$$

Also, define $f(\mathbf{E}, \mathbf{W}) := g(\mathbf{E}) + \langle \mathbf{W}, \mathbf{E} \rangle$. Our aim is to show that for all feasible perturbations $\mathbf{E}$, there exists $\mathbf{W}$ such that,

$$f(\mathbf{E}, \mathbf{W}) = g(\mathbf{E}) + \langle \mathbf{W}, \mathbf{E} \rangle > 0. \tag{3.7}$$

Note that $g(\mathbf{E})$ does not depend on $\mathbf{W}$.

We can use the following lemma from [1]:

**Lemma 3.1.** *Given* $\mathbf{E}$*, assume there exists* $\mathbf{W} \in \mathcal{M}_{\mathbf{U}}$ *with* $\|\mathbf{W}\| < 1$ *such that* $f(\mathbf{E}, \mathbf{W}) \geq 0$*. Then at least one of the followings holds:*

- *There exists* $\mathbf{W}^* \in \mathcal{M}_{\mathbf{U}}$ *with* $\|\mathbf{W}^*\| \leq 1$ *and* $f(\mathbf{E}, \mathbf{W}^*) > 0$.

- *For all* $\mathbf{W} \in \mathcal{M}_{\mathbf{U}}$, $\langle \mathbf{E}, \mathbf{W} \rangle = 0$.

*Proof.* Let $c = 1 - \|\mathbf{W}\|$. Assume $\langle \mathbf{E}, \mathbf{W}' \rangle \neq 0$ for some $\mathbf{W}' \in \mathcal{M}_{\mathbf{U}}$. If $\langle \mathbf{E}, \mathbf{W}' \rangle > 0$, choose $\mathbf{W}^* = \mathbf{W} + c\mathbf{W}'$. Otherwise, choose $\mathbf{W}^* = \mathbf{W} - c\mathbf{W}'$. Since $\|\mathbf{W}'\| \leq 1$, we have, $\|\mathbf{W}^*\| \leq 1$ and $\mathbf{W}^* \in \mathcal{M}_{\mathbf{U}}$. Consequently,

$$f(\mathbf{E}, \mathbf{W}^*) = f(\mathbf{E}, \mathbf{W}) + \langle \mathbf{E}, c\mathbf{W}' \rangle > f(\mathbf{E}, \mathbf{W}) \geq 0 \tag{3.8}$$

■

The following two steps are required to conclude that $(\mathbf{L}^0, \mathbf{S}^0)$ is the unique optimal with high probability:

1. Construct $\mathbf{W} \in \mathcal{M}_{\mathbf{U}}$ with $\|\mathbf{W}\| < 1$, such that $f(\mathbf{E}, \mathbf{W}) \geq 0$ for all feasible perturbations $\mathbf{E}$.

2. For all non-zero feasible $\mathbf{E} \in \mathcal{M}_{\mathbf{U}}^{\perp}$, show that $g(\mathbf{E}) > 0$.

The analysis in [1] relies on the independence of entries of $\mathbf{A}$ to use Bernstein-type concentration inequalities for the sum of independent random variables and the bound on spectral norm of random matrices with independent entries. We make the following observation: Split $\mathbf{A}$ filled via random triangle queries into three parts, $\mathbf{A} = \mathbf{A}_1 + \mathbf{A}_2 + \mathbf{A}_3$. For each triangle query, allocate one edge to each part randomly. If an edge gets queried as a part of multiple triangle queries, keep one of them randomly. Each $\mathbf{A}_i$ now contains independent entries. The edge density in $\mathbf{A}_i$ is $r_t p_T / 3$ and $r_t q_T / 3$ inside the clusters and outside respectively. This allows us to use the results on concentration of sum of independent random variables. So step 2 follows the same arguments from [1]. For the construction of dual certificate, we need to show that the candidate in the analysis of [1] still works, and provide suitable bound on its spectral norm accounting for the fact that we have dependent entries.

## 3.1 Dual Certificate Candidate

The dual certificate candidate is of the form:

$$\mathbf{W}_0 = \sum_{i=1}^{K} c_i \mathbb{1}^{n \times n}_{\mathcal{R}_{i,i}} + c \mathbb{1}^{n \times n} - \lambda \mathbb{1}^{n \times n}_{\mathcal{A}_1^c}, \tag{3.9}$$

where $\{c_i\}_{i=1}^K$, $c$ are real numbers to be determined and $\mathcal{A}_1$ refers to the locations of $\mathbf{A}$ that are 1. Set $\mathcal{S}^c$ is complement of the set $\mathcal{S}$. The notation $\mathbb{1}_{\mathcal{S}}^{n \times n}$ a matrix that has 1's in the entries corresponding the locations in set $\mathcal{S}$ and zero everywhere else. So, $\mathbb{1}_{\mathcal{A}_1^c}^{n \times n}$ has entries equal to 1 whenerver $\mathbf{A}$ has a zero in it. $\mathcal{R}_{i,i}$ denotes the positions in $\mathbf{A}$ that correspond to the nodes in cluster $i$, that is *inside* of the cluster $i$. $\mathbb{1}^{n \times n}$ denotes a matrix with all its entries equal to 1.

We split $\mathbf{W}_0$ into three parts:

$$\mathbf{W}_0 = \mathbf{W}_0^{(1)} + \mathbf{W}_0^{(2)} + \mathbf{W}_0^{(3)},$$

with

$$\mathbf{W}_0^{(i)} = \sum_{i=1}^K \frac{c_i}{3} \mathbb{1}_{\mathcal{R}_{i,i}}^{n \times n} + \frac{c}{3} \mathbb{1}^{n \times n} - \frac{\lambda}{3} \mathbb{1}_{(\mathcal{A}_i)_1^c}^{n \times n},$$

where $(\mathcal{A}_i)_1$ referes to the locations of $\mathbf{A}_i$ that are 1.

Expected value of an entry of $\mathbf{W}_0$ inside cluster region $\mathcal{R}_{i,i}$ is

$$\frac{c_i}{3} + \frac{c}{3} - \lambda(1 - r_t \frac{p_T}{3}),$$

and between the clusters is

$$\frac{c}{3} - \lambda \left(1 - r_t \frac{q_T}{3}\right).$$

To get zero mean entries, we set:

$$c = 3\lambda \left(1 - r_t \frac{q_T}{3}\right), \text{ and, } c_i = -\lambda r_t(p_T - q_T).$$

Thus the entries of $\mathbf{W}_0^{(i)}$ inside the cluster region $\mathcal{R}_{i,i}$ is $\lambda \left(1 - r_T \frac{p_T}{3}\right)$ w.p $r_t \frac{p_T}{3}$ and $-\lambda r_t \frac{p_T}{3}$ w.p $1 - r_t \frac{p_T}{3}$. Between the clusters, it is $\lambda(1 - r_t \frac{q_T}{3})$ w.p $\frac{q_T}{3}$ and $-\lambda r_t \frac{q_T}{3}$ w.p $1 - r_t \frac{q_T}{3}$. And the entries are all independent. Hence we can use the results on spectral norm of random matrices with mean zero entries that are independent. Note that the randomness in $\mathbf{W}_0^{(i)}$ is from due to the randomness in $\mathbf{A}_i$. Using the same arguments as in [1], we can conclude,

$$||\frac{1}{\lambda} \mathbf{W}_0^{(i)}|| \le 2\sqrt{n} \sqrt{r_t \frac{q_T}{3} (1 - r_t \frac{q_T}{3})} + 2\sqrt{n_{\max}} \sqrt{r_t \frac{q_T}{3}(1 - r_t \frac{q_T}{3}) + r_t \frac{p_T}{3}(1 - r_t \frac{p_T}{3})}$$

Since, $\mathbf{W}_0 = \mathbf{W}_0^{(1)} + \mathbf{W}_0^{(2)} + \mathbf{W}_0^{(3)}$, by triangle inequaly,

$$||\mathbf{W}_0|| \le \lambda 3 \left(2\sqrt{n} \sqrt{r_t \frac{q_T}{3}(1 - r_t \frac{q_T}{3})} + 2\sqrt{n_{\max}} \sqrt{r_t \frac{q_T}{3}(1 - r_t \frac{q_T}{3}) + r_t \frac{p_T}{3}(1 - r_t \frac{p_T}{3})}\right)$$

# References

[1] Ramya Korlakai Vinayak, Samet Oymak, and Babak Hassibi. Graph clustering with missing data: Convex algorithms and analysis. In *Neural Information Processing Systems Conference (NIPS)*, 2014.