[Reviews · NeurIPS 2016]

Reviewer 1

Summary

This paper studies the task of clustering a group of items via a crowdsourcing platform. In contrast to asking workers to label items, they are asked "comparison" queries and the collected information is used to form an adjacency graph among items. A clustering algorithm (based on low rank+sparse decomposition) is then applied to the resulting graph to recover the underlying clusters. Several generative models are proposed and under such models the cost of each query is defined as its entropy. On real datasets, where such models are not available, the costs are considered proportional to the number of items in the query. Fixing budget, it is shown empirically that the triangle queries gives better recovery performance of clusters than the edge queries. This is because they give more reliable edges (although dependent) and even more number of edges (effectively twice).

Qualitative Assessment

The results are presented in somewhat confusing manner. I do not find the stated results coherent and connected in a neat way. The model and the problem are interesting but I do not think the problem has been solved properly. Previously developed tools are used to obtain some partial results that fall short in providing a complete picture of the limits and tradeoffs for the problem of interest. Here are some suggestions to follow: 1) Fixing a budget B, stated how the performance of clustering behaves in terms of B for edge and triangle queries. Right now, theorem 1 concerns exact recovery and the required condition for that in terms of smallest and largest cluster sizes and within- ultra cluster probabilities. On the other hand, the main claim of the paper is that triangle queries outperform edge queries under fixed budget and ergo the main theorem should be focused in that direction. 2) How robust is the claim (dominance of triangle to edge queries) w.r.t choice of clustering algorithm? 3) There are several methods for clustering algorithm, why the formulation (4.1)? Especially that the nuclear norm optimization in the objective makes the problem computationally expensive. 4) What is the choice of lambda? (I guess it is something like 1/\sqrt{n} but it is not mentioned.) 5) In formulation (4.1) don't we need a constraint like A = L+S? 6) Under generative models, cost of a query is measured by its entropy. This should somehow depend on the previous queries and their outcomes, as it is a measure of new amount of information about the adjacency matrix by that query. However, in the current format such dependence is not clear.

Confidence in this Review

3-Expert (read the paper in detail, know the area, quite certain of my opinion)


Reviewer 2

Summary

The paper discusses the problem of discovering cluster structure of data via crowdsourcing edge-queries, or triangle queries, to human non-experts (such as amazon mechanical turkers). An error model for recovery of the cluster structure is discussed both for edge queries and triangle queries. It is argued, both theoretically and experimentally, that the triangle query model outperfroms the edge-query model. Also, a sufficient condition is shown for the minimum number of queries required to reconstruct the cluster structure, according to a particular reconstruction algorithm.

Qualitative Assessment

The paper studies an interesting problem. However, in my opinion, the overall work is rather incremental and the novelty is limited. The error models for edge and triangle queries are quite straightforward, they use parameters that are not easy to set, and make assumptions that are not well justified (i.e., the parameters p and q are the same for all triples). The main theoretical contribution, Theorem 1, is specific to one reconstruction algorithm, and it is not clear what is the impact of the theorem in practice. All in all, the paper is sound, and the result (that the triangle-query model is more effective than the edge-query model) is interesting, however, many arguments are hand waving, and the overall quality of the paper is not at a NIPS standard.

Confidence in this Review

2-Confident (read it all; understood it all reasonably well)


Reviewer 3

Summary

The paper answers a simple yet significant question: What types of queries are ideal when we intend to crowdsource a clustering task. The queries essentially ask workers about the similarity of items. Two types of queries are presented: (a) the edge query, and (b) the triangle query. The former randomly selects two pairs of items and asks if they are similar. The latter samples three items and asks how the three compare (e.g., all the same, all different, 1 and 2 similar). The paper argues that triangle queries are more fit for the clustering task. The performed experiments confirm this claim. The main technical contribution of the paper is to explain why this is the case. At a first glance, the claim is counter-intuitive as the edges in a triangle query are dependent. Thus an average edge in a triangle query should not be as informative as an edge in the edge query. While the previous observation is valid and certainly true, the analysis in the paper shows that the edges in the triangle query are more reliable (i.e. they have smaller error) exactly due to the dependency. It turns out that smaller error rate for edges in the triangle query is indeed more beneficial for the clusternig task. A fair comparison between the two type of queries can be carried out by choosing the number of triangle queries (T) and edge queries (E) so that they reveal the same number of edges (i.e., when E = 3 * T). While in this case, the triangle queries outperform the edge queries, the authors argue that the triangle queries are not three times as difficult edge queries. This claim is also supported by their experiments as well as their analysis. Overall, I think the paper is neat and straightforward. The message is very helpful for researchers working on crowdsourcing platforms.

Qualitative Assessment

I think the paper is well-written, concise and straight to the point. The technical results are sound and valid and the message is important for researchers who work on crowdsourcing. Also, I think the experiments support the claims that the authors put forward in the paper. I have a few concerns/comments which I list below. (1) In the beginning of section 2.3, the article says: "For simplicity, consider a graph with K clusters of size m". Is this only for the simplicity of the discussion? Or is this the only case for which the analysis goes through? This is not clear, and I strongly recommend the authors to highlight this. (2) I think the presentation can be improved. While the text reads well, it feels to me that in some places the introductory paragraphs are missing. For instance, section 2 suddenly talks about configurations which I was not certain what they represent. This requires the reader to do a couple of passes to make sure his guesses were right about what a configuration might be.

Confidence in this Review

2-Confident (read it all; understood it all reasonably well)


Reviewer 4

Summary

The paper considers the problem of clustering a partially observed graph and compares two types of queries: edge queries vs triangle queries. The problem is motivated by the task of clustering unlabeled items using crowdsourcing platforms. In this setting, an edge query reveals a random pair of items to a crowd worker for comparison whereas a triangle query reveals three random items and asks if any two of them, none of them, or all items are in the same cluster. The authors then try to answer the following question: for a fixed query budget, which query type can better retrieve the true underlying graph and therefore results in a more exact clustering? They provide a few generative stochastic models for both query types. The models allow them to compute the probability that a random edge in the observed graph is in error. They show that under reasonable assumptions on the parameters of their models, triangle queries find more reliable edges. They provide empirical evidence by presenting results from several simulations and two real data sets. Their experiments show that clustering via triangle queries outperforms clustering via edge queries by finding more reliable edges and finding more edges for a fixed budget.

Qualitative Assessment

The authors propose a novel strategy for partially observing a graph that consists of disjoint cliques(clusters). As they mention in the paper, an important application of this problem is in crowdsourcing platforms where a fixed budget is given and we want to cluster a large set of items as precisely as possible. While I think the problem is interesting and potentially useful across a variety of applications, I have two concerns regarding the novelty and thoroughness experimental analysis. Regarding novelty, although the authors provide some related work, I feel they could greatly improve on the comparison on two areas in particular: (1) the benefit of triadic comparisons in crowdsourcing and (2) the utility of sampling triangles for graph or adjacency matrix estimation. Regarding (1) the authors do mention related work but there is no experimental comparison with any of them. Similarly for (2)"Adaptively Learning the Crowd Kernel" provides a method to estimate the adjacency matrix -- how does this compare to the current work? Along this line the authors must either compare their approach to other approaches and give a clear explanation of why this is not possible. Further, the experiments do not leave me convinced that querying triangles is in general a better strategy. This is because of the way that authors compare costs of an edge query with a triangle query. In order to have a fair comparison between the number of edge queries and triangle queries that one can generate with a fixed budget, each query type must be associated with a cost. The authors suggest that the cost of a query can be measured by computing its entropy which is only available if the generative model of the graph is known. For real world problems, the authors use the average response time per query as the cost of a query. In section 3, they claim that empirical studies shows that the cost of a triangle query is approximately 1.5 times of an edge query, but there is no reference for this. Of course one can empirically estimate this ratio for a new data set by finding the average response time per query for each query type, but what if this ratio is very large and therefore only a few triangle queries are allowed with the given budget? It is not clear to me why querying triangles is a better choice in this scenario. On the other hand, the paper claims that the cost of a triangle query is never more than 3 times the cost of an edge query because each triangle query contains three edges. I am not convinced by this reasoning. It seems that they ignore the fact that identifying dependencies between edges makes the problem harder to solve but at the same time they are using the advantage of solving triangle queries which are more reliable than edge queries exactly because they consider edge dependencies. I would suggest the authors compare the performance of edge queries vs triangle queries over a wider range of cost ratios to study if there is a turning point such that using edge queries reveals more information about the graph for a fixed budget(i.e., the cost of triangle queries are so high that we can only make a few of them). Followed by an empirical study that shows triangle queries are statistically unlikely to be so expensive.

Confidence in this Review

2-Confident (read it all; understood it all reasonably well)


Reviewer 5

Summary

This paper proposed to conduct edge query and triangle query in crowdsourcing clustering. The main contributions are: 1) the model description 2) random triangle query will recover the true cluster with high probability, 3) empirical validation.

Qualitative Assessment

If the edge/triangle query are not proposed in crowdsourcing in previous work, I think this kind of methods is reasonable for crowdsourcing. The comparison is much easier for human. My concerns/suggestions are as follows: (1) This paper is not well-written. There are many gramma errors, and notations without explanations. For example, “Even though,..” in abstract; T_B, H_\triangle, etc. (2) the structure of the paper is not smooth nor compact. There are no transition sentences between sections, or overall sentences in each sections. Readers are very easy to get lost. (3) The technique contributions are not significant. To be specific, I don't’ think there should be one individual section for Value of a Query; the result in Theorem 1 is not strong without indicating the probability of the recovery; how about the recovery probability for edge query? (3) the measurement of the efficiency of the new method is not specific or convincing. In the experiments, the authors claim triangle query is better since its adjacency matrix with significantly less confusion. Is there a clear measurement?

Confidence in this Review

2-Confident (read it all; understood it all reasonably well)


Reviewer 6

Summary

This paper tries to answer whether edge queries or triangle queries are more suitable for crowdsourced clustering. To this end, it firstly introduced the two different kinds of labeling interfaces, and then the one-coin model and the stochastic block model are discussed to define the confusion matrix of labeling. A convex optimization problem with theoretical guarantee is investigated to recover the true adjacency matrix from the partial observations. Finally, both theoretical and empirical results demonstrate that the triangle interface is significantly better than then the edge interface.

Qualitative Assessment

I think the motivation of this paper is very interesting. Intuitively, displaying more instances in a same interface can accelerate the labeling procedure. Many papers, including crowdclustering, are benefitted from this idea. However, as far as I can see, none of them has given a theoretical analysis to prove that this idea really works. The triangle interface discussed in this paper can be viewed as a special case of the interface used in crowdclustering when there are 3 items in each task. According to the paper, both theoretical and empirical results demonstrate its priority over the edge interface. My main concern about this paper is that I think it lacks of in-depth analysis, the contribution is okay but not enough good for NIPS. For example, the ability and bias of labelers have great impact on the crowdsourcing quality, but they are not discussed in this paper. And what if we display more than 3 items in a same task? Will it lead to even better efficacy for labeling? Besides, the presentation of this paper needs further polishing since I find many typos. For example, I think the A_{i,j} in Eq. (4.1) should be A_{i,j}^obj, which is consistent with the referred paper [16]. The referred paper [11] and [17] do not have titles in the reference list. Moreover, some figures are not well scaled.

Confidence in this Review

2-Confident (read it all; understood it all reasonably well)